# Preprocessing Variability in fMRI Predictive Modeling: Effects of Distortion Correction on Functional Connectivity-Based Predictions

Zishen Li[1], Bishal Lamichhane[1], Ankit Patel[1,2], Ramiro Salas[2], Nidal Moukaddam[2], and Ashutosh Sabharwal[1]

*Abstract*—**Functional connectivity (FC)-based predictive modeling is a widely used approach in resting-state fMRI studies to predict various mental states such as attention, depression, and impulsivity. FC-based predictive modeling often employs a standard procedure: parcellating the brain into regions of interest (ROIs) using a predefined atlas, computing ROI-to-ROI FC, and applying predictive models to estimate behavioral or clinical measures. However, many existing studies focus solely on end-to-end prediction performance and often overlook the influence of preprocessing choices on FC features and downstream predictive model performance. Assessing the preprocessing effect is crucial because it can significantly influence the spatial accuracy of ROI partition, FC measures, and predictive modeling performance, potentially reducing reproducibility. In this study, we investigated the impact of fMRI preprocessing strategies, particularly fieldmap distortion correction, on the resulting FC features and the performance of machine learning models predicting sensation-seeking. We compared two preprocessing pipelines: with distortion correction (*DC*) and without (*NDC*). FC matrices were computed from each pipeline and used to train machine learning models to predict sensation-seeking trait. We showed that different preprocessing choices can lead to substantial differences in FC values and model predictions. The prediction model trained on *DC* data achieved $R^2$ of 0.34, while the model trained on *NDC* data has a lower $R^2$ of 0.21. Moreover, we observed notable differences in the key predictive connections between the *DC* and *NDC* pipelines, involving the brain regions such as cerebellum, prefrontal cortex, cingulate cortex, and subcortical regions, which also showed the largest voxel shifts following distortion correction. Our findings revealed the important role that preprocessing strategies play in functional connectivity-based modeling and raised the important issue of accounting for preprocessing variability in fMRI predictive modeling.**

*Index Terms*—**rs-fMRI, Functional Connectivity, Preprocessing, Distortion Correction, Predictive Modeling**

## I. INTRODUCTION

Functional magnetic resonance imaging (fMRI) is a non-invasive neuroimaging technique widely used to study brain activity and its relationship with human health. A key feature of fMRI is functional connectivity (FC), commonly defined by the temporal correlation between brain regions, representing intrinsic brain function. FC is linked to various physical and psychological processes in the human body, making it an essential feature in studying brain-behavior relationships [1].

However, fMRI data acquisition is often heterogeneous due to differences in scanning devices and protocols, and inter-subject variability. Therefore, preprocessing is important to mitigate fMRI heterogeneity, reduce artifacts, and standardize the data [2]. Although most studies typically follow similar steps, which include motion correction, coregistration, signal denoising, filtering, and spatial normalization, the implementation details often differ. There is strong evidence that signal filtering, white matter/cerebrospinal fluid regression, and global signal regression influence the estimation of the brain connectome [3]–[5]. However, one critical step in preprocessing, distortion correction, is less examined [6]. This step corrects geometric distortions from magnetic field inhomogeneity using a fieldmap, which represents variations in the static magnetic field [7]. Distortion correction improves the alignment between the functional and the structural images and reduces mislocalization after spatial normalization to standard space. However, fieldmap acquisition requires additional scan time and is often unavailable in many publicly available fMRI datasets that were collected and released years ago [8]. Thus, many studies overlook this step, especially when the areas of interest may not be deemed particularly vulnerable to distortion by the study team [6]. To our knowledge, the impact of distortion correction on the downstream connectivity-based predictive modeling has not been examined. We hypothesize that the inclusion or exclusion of fieldmap distortion correction affects functional connectivity measures, potentially influencing the outcomes of machine learning models trained to predict behavioral traits from these features.

To address the above hypothesis, we consider the case of predicting sensation-seeking trait, a sub-dimension of impulsivity, using functional connectivity as input features. Recent studies have employed machine learning approaches for similar predictions [9]–[11]. However, those studies primarily focused on the performance metric without considering model sensitivity. This is important because, given the high-dimensionality nature of FC data, even small variations can lead to significant changes in model outcomes. Prior research has demonstrated that different preprocessing can lead to systematically different FC matrices [3], [4]. The main gap lies in understanding how preprocessing choices influence downstream predictive modeling, which can limit the reproducibility and interpretability of FC-based predictions in neuroimaging.

In this study, we aim to bridge this gap by evaluating how distortion correction affects both functional connectivity (FC) estimation and machine learning-based behavioral prediction. We implemented two fMRI preprocessing pipelines, one with

---

[1] Department of Electrical and Computer Engineering, Rice University, Houston, TX, USA

[2] Baylor College of Medicine, Houston, TX, USA

distortion correction (*DC*) versus one without distortion correction (*NDC*, no distortion correction), and computed FC features from each. The computed features were then used to train a machine-learning model to predict sensation-seeking scores. We evaluated this modeling on the 128 male participants from the MPI Leipzig Mind-Brain-Body (LEMON) dataset to avoid any gender confounders. The male subset was selected for its larger sample size and greater variability in sensation-seeking scores compared to the female population in the dataset.

We observed significant subject-level differences in functional connectivity ($p < 0.05$) derived from the two preprocessing pipelines in 32% of the subjects. The most affected connections involved the cerebellum, vermis, prefrontal cortex, cingulate cortex, subcortical regions (e.g., amygdala, hippocampus), and parietal lobe (e.g., precuneus, angular gyrus) that also exhibited the highest voxel shift after distortion correction. Our findings further demonstrated that FC-based sensation-seeking prediction varied substantially depending on whether distortion correction was applied. The prediction model trained on *DC* data achieved $R^2$ of 0.34, while the model trained on *NDC* data had a lower $R^2$ of 0.21. This study highlighted the importance of evaluating the effects of preprocessing choices on the reliability and generalizability of machine learning predictive models in neuroimaging.

## II. MATERIALS AND METHODS

### A. Data

*1) Participants:* The dataset used in this study is part of the large MPI Leipzig Mind-Brain-Body (LEMON) dataset, which comprises multi-modal data, including brain imaging data (MRI and EEG), cognitive assessments, emotional measures, and peripheral physiology data for studying the relationships between mind, brain, and body functions [12]. The 205 participants fell into two age groups: 139 from a younger group aged $20-35$, and 66 from an older group aged $59-77$.

*2) Self-reported sensation-seeking score:* Sensation-seeking, a sub-dimension of impulsivity, reflects a preference for stimulation and excitement. In the LEMON dataset, it's measured using the UPPS impulsivity questionnaire [13]. UPPS covers four sub-dimensions of impulsivity: Urgency, Lack of Premeditation, Lack of Perseverance, and Sensation-Seeking. Participants rated 45 questionnaire items on a 4-point scale (higher score indicates stronger trait). Each item relates to one of these sub-dimensions, and the aggregated scores reflect the impulsivity level across the sub-dimensions. Figure 1 shows the distribution of age and sensation seeking score across the male and female groups in the LEMON dataset.

### B. Methods

*1) Preprocessing Pipelines:* In the LEMON dataset, rs-fMRI and structural MRI scans were obtained using a 3 Tesla scanner. Participants were instructed to stay awake with their eyes open during rs-fMRI acquisition. We employed two preprocessing pipelines (Figure 2), with and without fieldmap distortion correction, referred to as *DC* and *NDC*.

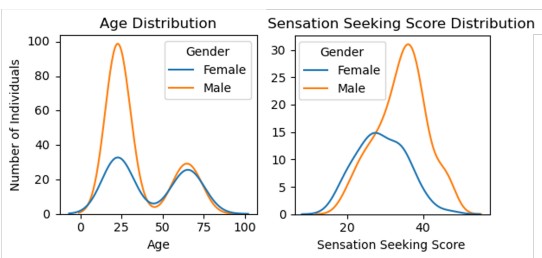

Fig. 1. Age and sensation seeking score distribution across males and females.

Aside from fieldmap distortion correction, other steps were identical for *DC* and *NDC* pipelines: 1) removing the first 5 slices to ensure stable signals; 2) 3D motion correction; 3) rigid-body coregistration to anatomical image; 4) signal denoising; 5) band-pass filtering; 6) mean centering and variance normalization; 7) spatial normalization to MNI152 2mm standard space. For *DC*, fieldmap correction was performed before coregistration to the anatomical image. Fieldmap imaging was acquired with a double-echo gradient echo sequence that included a magnitude image and a phase image. FSL FUGUE [14] was used to compute and correct the spatial shift of each voxel from the original functional image space due to magnetic inhomogeneity. After preprocessing, time series extracted from each brain region were used to construct functional brain connectivity networks. After quality control, we excluded 3 from the initial 131 male subjects, leaving 128 subjects for subsequent analysis.

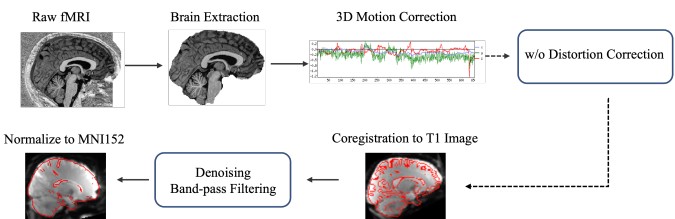

Fig. 2. The workflow of fMRI preprocessing. We compared two preprocessing pipelines, with and without distortion correction; other steps remain identical.

*2) Brain functional connectivity:* fMRI measures brain activity by detecting changes in blood flow and oxygenated hemoglobin, known as blood oxygen level-dependent (BOLD) signals. Temporal correlations of the BOLD signal between pairs of brain regions are used to represent brain functional connectivity (FC). As illustrated in Figure 3 (a), after preprocessed through *DC* and *NDC* pipelines, respectively, we partitioned the voxel-level fMRI data into 116 regions of interest (ROIs) with Automated Anatomical Labelling (AAL) Atlas [15] to reduce the high dimensionality. Next, we extracted ROI-wise time series by averaging BOLD signals across voxels within each ROI. FC was computed as the Pearson correlation between the time series of all ROI pairs. This yields 6670 FCs for each individual, representing the brain connectivity network. Subsequently, we used the absolute correlation value to represent the connection strength,

which was the input feature for the predictive model.

*3) Assessment of the influence of distortion correction on FC:* We evaluated the influence of fieldmap distortion on FC with the following measurements:

- Degrees of Voxel Shift after Distortion Correction: We calculated the spatial shift of each voxel from the original functional image space after distortion correction.
- Distribution of Functional Connectivity: After calculating FC from *DC* and *NDC*, we reported the global difference between the two FC with mean and standard deviation. We also conducted a statistical test to compare the distribution of FC at the subject level.

*4) Sensation-seeking score prediction framework:* As a **baseline**, we implemented the Elastic Net model from previous literature [10], the only work predicting sensation-seeking from rs-fMRI to our knowledge. We applied this model to all 128 male participants from the LEMON dataset.

Our predictive model employed nested cross-validation to predict sensation-seeking scores from functional connectivity, as illustrated in Figure 3 (b). This consists of two loops: an inner loop for feature selection and hyperparameter tuning, and an outer loop for unbiased model performance evaluation [16]. Within each outer training set, we performed 10-fold cross-validation as the inner loop to optimize model configuration and feature selection. We adopted leave-one-subject-out cross-validation for the outer loop to evaluate the model's predictive performance and generalization ability across each individual. We evaluated predictive performance using Pearson's correlation coefficient (*r*), $R^2$, Root Mean Squared Error (*RMSE*), and normalized *RMSE* (*NRMSE*).

For the predictive model, we evaluated a set of machine-learning models to capture either linear or nonlinear relationships between FC and sensation-seeking, including Extreme Gradient Boosting (XGBoost) and Generalized Linear Model (GLM) - Elastic Net and Lasso. These models are widely used to handle high-dimensional datasets and prevent overfitting. We first applied Light Gradient Boosting Machine (Light-GBM) for feature selection to reduce dimensionality and keep the most informative features for prediction, selecting top-k features that achieved the highest average cross-validated $R^2$ in the inner loop. The optimal number of features was then applied to the held-out fold in the outer cross-validation. We optimized model settings through inner cross-validation, including tree depth and feature split thresholds for XGBoost, the regularization term for ElasticNet, and Lasso. Feature selection and hyperparameter tuning were performed exclusively on training data within each fold to avoid data leakage.

Previous work found that the associations between FC and sensation-seeking were not uniform across ages and proposed an age-specific modeling approach, which outperformed the single all-ages model [11]. Building on the age-specific modeling approach, we partitioned the male population into different age groups and implemented the prediction framework within each age group. We focused on the male population due to its higher variability in sensation-seeking scores and a larger sample size in each age group (as illustrated in Figure 1).

To ensure sufficient data samples in each group, we merged the 30–34 ($N = 5$) and 35–39 ($N = 1$) groups with 25–29, and combined older participants into a 55–79 group. The final age groups were 20–24 ($N = 46$), 25–39 ($N = 50$), and 55–79 ($N = 32$). As the number of female participants in key age ranges was insufficient for reliable modeling (e.g., $N = 22$ for 20–24 and $N = 21$ for 25–39), we focus on the male subset to avoid any gender-related confounding.

*5) Model Sensitivity:* Due to the high dimensionality of functional connectivity data, even small variations in the data can lead to substantial perturbations in model performance. To evaluate the sensitivity of our prediction framework to such variations, we added random Gaussian noise with varying variance to the FC features and reran the prediction pipeline. We calculate the $R^2$ for each noise level, showing the influence of FC variations on model predictive accuracy.

## III. RESULTS

### A. Assessment of the influence of distortion correction on FC

*1) Degrees of Voxel Shift after Distortion Correction:* We quantified the voxel-wise spatial shift using the unwarping shift map derived from the field map images. Figure 4 shows the spatial shift across all voxels and a heatmap of the normalized voxel shift magnitude throughout the brain after applying distortion correction. The mean voxel shift is 1.18*mm*. Notably, the most substantial shifts occurred in brain regions located near the edges, including the vermis ($6.32 \pm 0.65mm$), cerebellum ($5.30 \pm 1.37mm$), medial orbitofrontal gyrus ($5.41 \pm 0.02mm$), anterior cingulate gyrus ($4.3 \pm 0.3mm$), superior and middle temporal pole ($3.66 \pm 1.51mm$), precuneus ($4.53 \pm 0.03mm$) amygdala ($5.37 \pm 0.03mm$), caudate nucleus ($4.53 \pm 0.16mm$), hippocampus ($5.15 \pm 0.14mm$), and parahippocampal gyrus ($6.61 \pm 0.23mm$) areas. These regions have also been demonstrated to be affected by distortion in previous studies [6], [17], [18].

*2) Distribution of Functional Connectivity:* We compared the distribution of functional connectivity derived from the pipelines with distortion correction (*DC*) and without distortion correction (*NDC*). The overall distribution of FC values across all subjects appears similar between the two pipelines (*DC*: $0.31 \pm 0.26$, *NDC*: $0.32 \pm 0.26$). For subject-level differences in FC between the two pipelines, a paired t-test indicated a significant preprocessing effect in 32% of the subjects ($p < 0.05$), showing that distortion correction can substantially alter FC estimates at the individual level. Although overall FC distribution differences were subtle, these subject-level variations may lead to meaningful differences in model performance. Figure 5 shows brain connections with significant FC differences ($p < 0.05$) between pipelines and highlights the involved brain regions. Notably, these brain regions with the most significant FC differences largely overlap with areas exhibiting large voxel shifts in Figure 4, including regions such as the cerebellum, vermis, prefrontal regions, cingulate gyrus, subcortical regions (e.g., amygdala, hippocampus, parahippocampal gyrus), and parietal lobe (e.g.,

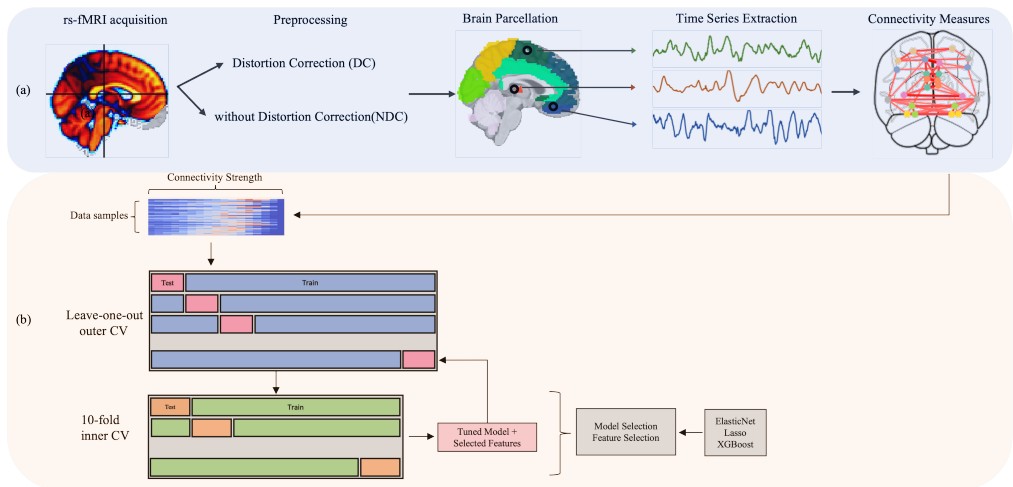

Fig. 3. Sensation-seeking score prediction framework: (a) The functional connectivity feature was constructed from two preprocessing pipelines, with and without distortion correction; (b) The functional connectivities then served as input features of the prediction framework to predict sensation-seeking scores.

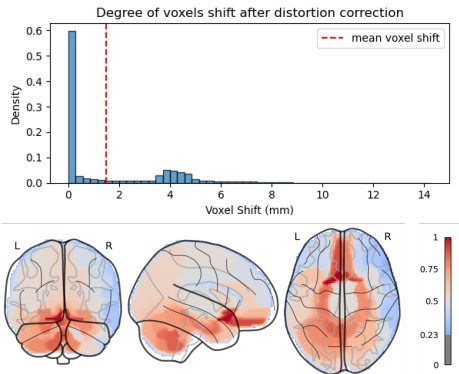

Fig. 4. Top: voxel shift after distortion correction. Mean voxel shift is $1.18mm$. Bottom: normalized voxel shift magnitude throughout the brain regions.

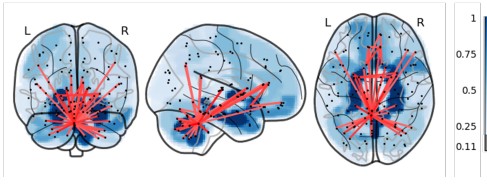

Fig. 5. Functional connectivity changes with distortion correction. Blue-white color represents the normalized magnitude of FC changes in brain regions.

precuneus, angular gyrus). These findings highlight the importance of correcting distortion for accurate FC estimation in regions susceptible to magnetic field inhomogeneities.

### B. Predictive performance

*1) Predictive performance on sensation-seeking prediction:* As described in the previous section, the results were obtained using outer leave-one-subject-out cross-validation within the nested framework.

As **baseline**, we fitted an ElasticNet model on the male group for *DC* and *NDC* pipelines. The result shows a notable difference in the prediction performance with the two pipelines (*DC*: $R^2 = -0.03, r = 0.07$, *NDC*: $R^2 = 0.036, r = 0.26$).

Building on the age-specific modeling [11], we segmented the dataset into age subgroups and developed predictive models tailored to each subgroup. Table I presents the evaluation metrics of *DC* and *NDC* within each subgroup, and Figure 6 illustrates the prediction results within each subgroup. The results show a significant difference in prediction performance between *DC* and *NDC*. With distortion correction, the model achieved high prediction performance in age groups $20-24$ ($R^2 = 0.21$) and $55-79$ ($R^2 = 0.37$) but achieved relatively low accuracy in the $25-39$ age group ($R^2 = 0.03$). While within the *NDC* group, we observed a significant drop in $R^2$ and $r$ in the $20-24$ ($R^2 = 0.01$) and $55-79$ ($R^2 = 0.0$), where *DC* achieved higher accuracy.

Figure 7 shows the aggregated prediction from all age groups. Similar to the results within each age group, *DC* group achieved higher accuracy ($R^2 = 0.34$) compared to the *NDC* group ($R^2 = 0.21$). Both *DC* and *NDC* achieved higher accuracy with age-specific modeling compared to the baseline (*DC*: $R^2 = -0.03$, *NDC*: $R^2 = 0.036$), which is in line with previous findings [11].

*2) Functional connectivity associated with sensation-seeking:* Our analysis revealed distinct associations between functional connectivity and sensation-seeking across different age groups. To identify strong predictors for sensation-seeking, we determined the connectivities that were consistently selected by over 80% of the cross-validation iterations. These robust predictors potentially provide valuable insights into the neural correlates of sensation-seeking. Figure 8 visualizes the involved brain regions in each group in both the distortion-corrected (*DC*) and non-distortion-corrected (*NDC*) pipelines.

In the age group 20–24, the *DC* pipeline highlighted key connections involving the prefrontal cortex (PFC), orbitofrontal cortex (OFC), temporal pole, cingulate cortex, and cerebellum. The *NDC* pipeline identified similar regions,

TABLE I
EVALUATION METRIC OF THE MODEL FITTED IN EACH SUBGROUP (FC = FUNCTIONAL CONNECTIVITY)

| Participants (N: number of individuals) | $r$ DC / NDC | $R^2$ DC / NDC | RMSE DC / NDC | NRMSE DC / NDC |
|---|---|---|---|---|
| Male aged 20-24 (N=46) | 0.47 / 0.11 | 0.21 / 0.01 | 5.45 / 6.10 | 20.9 / 23.4 |
| Male aged 25-39 (N=50) | 0.22 / 0.23 | 0.03 / 0.05 | 5.82 / 5.77 | 22.4 / 22.2 |
| Male aged 55-79 (N=32) | 0.62 / 0.18 | 0.37 / 0.0 | 4.62 / 5.86 | 23.1 / 29.3 |
| Male population (N=128) | 0.59 / 0.46 | 0.34 / 0.21 | 5.41 / 5.91 | 18.6 / 20.4 |

cessing pipelines were largely similar, the prediction results differed significantly. The differences in predictive performance between the *DC* and *NDC* pipelines may be attributed to variations in the derived FC features. As shown in Figure 5, regions such as the cerebellum, medial orbitofrontal cortex, cingulate cortex, and subcortical areas (amygdala, thalamus, hippocampus)-which exhibited the most significant FC changes after distortion correction- were also involved in the key connections for sensation-seeking. This overlap highlights that distortion correction can critically impact the identification of meaningful brain-behavior relationships in FC-based predictive modeling.

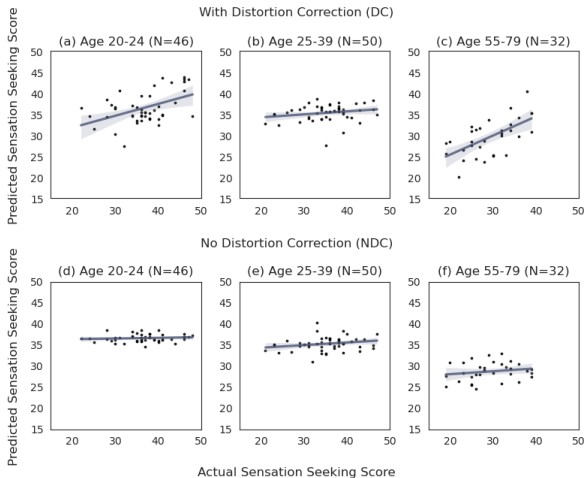

Fig. 6. Prediction result within each age group of *DC* and *NDC*. Top: Prediction with distortion correction. a) age group $20 - 24$ ($R^2 = 0.21$). b) age group $25 - 39$ ($R^2 = 0.03$). c) age group $55 - 79$ ($R^2 = 0.37$). Bottom: Prediction without fieldmap correction. d) age group $20 - 24$ ($R^2 = 0.01$). e) age group $25 - 39$ ($R^2 = 0.05$). f) age group $55 - 79$ ($R^2 = 0.0$).

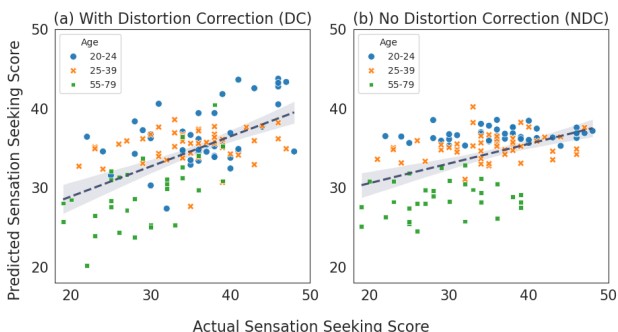

Fig. 7. Prediction result of the male population aggregated all age groups ($N = 128$). a) shows the prediction with fieldmap correction ($R^2 = 0.34$). b) shows the prediction without fieldmap correction ($R^2 = 0.21$)

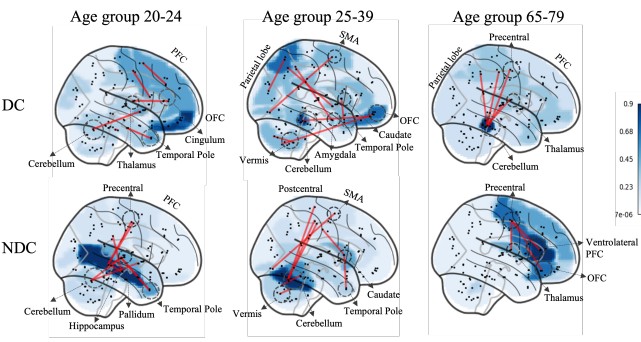

Fig. 8. Key brain regions associated with sensation-seeking within each age group. White-blue color shows the normalized sum of connections within brain regions. The brain regions involved are the cerebellum, vermis, frontal lobe (PFC, precentral gyrus), subcortical areas (caudate, amygdala, thalamus, hippocampus), and parietal lobe (precuneus, angular gyrus), which overlap with the most significant FC changes after distortion correction.

### C. Complementary evaluations - Alternate downstream prediction tasks and preprocessing choices

To provide a more thorough analysis of the impact of preprocessing on predictive performance, we evaluated two additional targets: 1) age and 2) attentional performance from the TAP-incompatibility test [19], which measures interference control towards incompatible stimuli as a behavioral outcome. For age prediction, both the distortion-corrected (DC) and non-corrected (NDC) pipelines performed well ($R^2 = 0.52$). For TAP scores, the *DC* pipeline outperformed *NDC* ($R^2 = 0.26$ vs. $R^2 = 0.15$), which is similar to our findings for sensation-seeking, showing a substantial performance gain from distortion correction.

Moreover, we also examined two debated denoising choices—global signal regression (GSR) and white mat-

including the cerebellum and temporal pole, with stronger connectivity in the hippocampus. For the 25–39 group, both pipelines showed key regions such as the cerebellum, temporal pole, supplementary motor area (SMA), and subcortical structures (e.g., amygdala, caudate). In the 55–79 group, the *DC* pipeline identified key connections between the cerebellum and precentral, parietal lobes, while *NDC* identified connections in the precentral gyrus, ventrolateral PFC, and OFC.

Although the brain regions involved in the two prepro-

ter/cerebrospinal fluid (WM-CSF) regression, using the distortion-corrected (DC) data as the baseline. For GSR, regressing out global signal confounds achieved $R^2 = 0.18$. In the WM-CSF regression, we regressed out WM-CSF confounders derived from the raw signal, instead of using CompCor denoising [20] in the *DC* pipeline, which leads to $R^2 = 0.10$. These analyses further demonstrate that preprocessing choices beyond distortion correction also lead to large impacts on prediction performance.

### D. Model sensitivity analysis

We conducted a model sensitivity analysis by adding Gaussian noise with different variances to the FC data. For each noise level, we computed the $R^2$ between predicted and actual sensation-seeking scores. As shown in Figure 9, model performance dropped substantially as noise increased. Given the high dimensionality of FC data, even small variations can lead to substantial changes in prediction accuracy. This also helps explain why differences in preprocessing strategies can impact the predictive model outcomes.

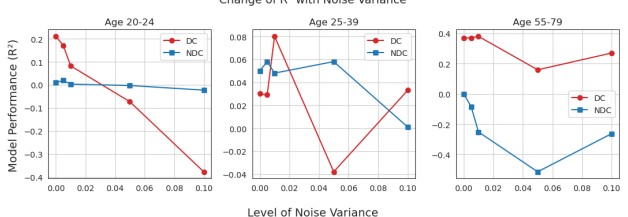

Fig. 9. Model sensitivity analysis shows how $R^2$ value drops as increasing noise was added in *DC* and *NDC* prediction ($\sigma^2 = 0.005, 0.01, 0.05, 0.10$)

## IV. DISCUSSION

### A. The impact of preprocessing in fMRI

In this study, we examined the influence of fMRI preprocessing on functional connectivity (FC) and its downstream impact on the prediction of sensation-seeking scores in 128 healthy male participants from the LEMON dataset.

Our predictive model achieved an $R^2$ of 0.34 and a root mean squared error (*RMSE*) of 5.41 when using the distortion-corrected (*DC*) pipeline, demonstrating that sensation-seeking can be predicted from resting-state fMRI with reasonable accuracy. In contrast, the model trained on data without fieldmap distortion correction (*NDC*) showed a reduced $R^2$ of 0.21. One likely explanation is that distortion correction significantly affects spatial registration, especially when ROI partitions are derived from a standard template-based atlas. Misalignment between fMRI and ROI template can lead to inaccurate signal extraction and functional connectivity estimates.

The selected functional connectivities associated with sensation-seeking involved brain regions such as the cerebellum, frontal lobe (e.g., prefrontal cortex, precentral), cingulate cortex, subcortical structures (e.g., amygdala, hippocampus, thalamus, caudate, pallidum), parietal lobe (e.g., angular gyrus, precuneus), and temporal pole, as shown in Figure 8. Previous

studies have shown that the cerebellum, thalamus, and hippocampus are involved in reward processing, which relates to sensation seeking [21], [22]. The prefrontal cortex plays a key role in impulse control and novelty detection. Its interaction with subcortical regions like the amygdala and hippocampus is linked to novelty-seeking [23]. The anterior cingulate cortex and orbitofrontal cortex are also involved in response inhibition, which relates to sensation-seeking [24]. Notably, many of these important regions identified in the prediction model, such as the cerebellum, medial orbitofrontal cortex, cingulate cortex, and subcortical regions (caudate, amygdala, hippocampus, thalamus), were also most affected by distortion correction (Figure 4). This overlap suggests the reason why distortion correction has a meaningful impact on FC measures and model performance.

Accurate functional connectivity estimation is essential for reliable neuroimaging biomarkers in clinical research. Our study showed that preprocessing choice, especially fieldmap correction, significantly impacts functional connectivity (FC) measures. Moreover, we demonstrated that these variations in FC due to preprocessing significantly influenced the performance of connectivity-based predictive modeling, with the case of sensation-seeking prediction. Although previous studies have explored the relationship between FC and sensation-seeking, they typically focused on end-to-end prediction performance while overlooking how preprocessing steps might affect the input features and consequently affect the model outcome. Our study emphasized the need to account for preprocessing variability when developing predictive models based on fMRI data, in order to enhance the reproducibility and clinical utility of fMRI-based models.

Our analysis also showed that the impact of preprocessing could depend on the downstream prediction target, as seen in the differing effects for age and TAP test prediction. This could be due to the neurobiological feature implicated in predicting the target. The predictive features for the TAP task involved regions such as the cerebellum, hippocampus, and amygdala, which are affected by distortion correction; while age prediction involved broader brain networks and global brain connectivity change [25], which are less sensitive to local changes with fieldmap distortion correction.

Fieldmap distortion correction mitigates spatial misalignment and benefits the signal extraction, especially when targeted brain regions are vulnerable to magnetic field distortion. We suggest that researchers incorporate fieldmap distortion correction in fMRI preprocessing if fieldmap data is available in the dataset. If fieldmap data is not provided, alternative distortion correction methods, such as field-reversed DTI or nonlinear registration using ANTs, can be considered [26]. We also showed that other preprocessing steps, such as global signal regression and WM-CSF regression, can also have a substantial impact on FC measures, which aligns with previous findings [3], [4]. While there is no clear consensus on the most optimal preprocessing choice, our study showed that variations in FC measures due to preprocessing can lead to significant differences in model outcomes. We suggest that researchers

should be aware of these effects as a potential source of variability when building predictive models.

### B. Limitation and future directions

One limitation of our study is the lack of investigation of prediction in the female subgroup due to the small number of female participants in each age group. To minimize potential gender-related confounding effects, we restricted our analysis to the male cohort as males exhibit a higher sensation-seeking score with greater variability, which is of higher clinical relevance [27]. Future research could aim to include a larger and more gender-balanced dataset to validate the findings in a larger population. We will also conduct a validation study on external datasets in future work to validate the generalizability of our findings across different samples, brain atlases, preprocessing pipelines, and prediction tasks, and provide a comprehensive investigation into the impact of preprocessing on functional connectivity modeling.

## V. CONCLUSIONS

Our study explored the influence of fMRI preprocessing choice and distortion correction on the derived functional connectivity features and downstream predictive modeling performance. By comparing two preprocessing pipelines—one with distortion correction (*DC*) and one without (*NDC*), we show that distortion correction affects the spatial alignment of fMRI and therefore impacts functional connectivity measures. The prediction model trained on *DC* data achieved $R^2$ of 0.34, while the model trained on *NDC* data has a lower $R^2$ of 0.21, demonstrating the sensitivity of FC predictive modeling to variations in preprocessing steps. Our findings pointed out the limitations of end-to-end FC-based predictive modeling and demonstrated the importance of accounting for preprocessing variability when interpreting functional connectivity-based predictions. Future research could extend this finding to a larger population or incorporate other processing choices to provide a comprehensive understanding of the critical role of preprocessing in the reproducibility and interpretability of functional connectivity-based predictive modeling.

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
