# OpenReview forum: "Preprocessing Variability in fMRI Predictive Modeling: Effects of Distortion Correction on Functional Connectivity-Based Predictions"
_IEEE.org/EMBS/BHI/2025/Conference — BHI 2025_

### Official Review · Reviewer_MUnh · 2025-07-09
**Review - Preprocessing Variability in fMRI Predictive Modeling: Effects of Distortion Correction on Functional Connectivity-Based Predictions**

**Confidence:** 3
**Clarity Of Writing:** good
**Clinical Significance:** fair
**Methodological Novelty:** good
**Overall Rating:** 6
**Final Rating:** 7

**Experiments And Results:**

fair

**Questions For The Authors:**

- Did you explore how other preprocessing steps (e.g., global signal regression, motion scrubbing) might interact with distortion correction to affect FC and prediction?

- Could your findings generalize to task-based fMRI or other behavioral phenotypes? Have you considered validating your approach on additional traits or datasets?

- Given the exclusive focus on male participants, do you expect similar improvements in model performance for mixed-gender or female-only cohorts?

- How stable are your model weights or feature importances across the LOOCV folds? Do the same regions/networks consistently contribute to prediction?

- Do you have recommendations for researchers using datasets that lack fieldmaps—e.g., is simulating distortion artifacts and testing model robustness a viable strategy?

**Strengths:**

This paper addresses an underexplored yet critical issue in fMRI-based machine learning: the impact of preprocessing variability on downstream predictions. By isolating the role of fieldmap distortion correction, the study offers novel insights into how spatial misalignment affects functional connectivity and behavior prediction. The use of a well-documented public dataset (LEMON) and robust cross-validation (nested LOOCV) adds credibility. The predictive framework is thoughtfully designed, with multiple age-specific models and sensitivity analyses using added Gaussian noise. Results are clearly reported, with visualizations linking voxel shifts to FC variation and prediction outcomes. Importantly, the study offers practical recommendations for researchers dealing with heterogeneous preprocessing pipelines, especially in legacy or public datasets lacking fieldmaps.

**Summary Of The Paper:**

This paper investigates how fMRI preprocessing decisions—specifically fieldmap-based distortion correction—impact functional connectivity (FC) measures and the performance of predictive models trained to estimate sensation-seeking behavior. Using resting-state fMRI data from 128 male participants in the LEMON dataset, the authors compared two preprocessing pipelines: one with distortion correction (DC) and one without (NDC). They computed FC matrices from both and used them to train machine learning models (e.g., XGBoost, Elastic Net) for sensation-seeking prediction. Results showed that DC significantly improved prediction performance (R² = 0.34) compared to NDC (R² = 0.21). Moreover, brain regions showing the largest voxel shifts due to distortion correction also exhibited the most significant FC differences and were crucial to prediction performance. The paper emphasizes the importance of accounting for preprocessing choices to enhance reproducibility in FC-based predictive modeling.

**Weaknesses:**

One major limitation is the exclusive focus on male participants, which reduces the generalizability of the findings, especially given known gender differences in impulsivity traits. The sample size (N=128) is moderate, but subgroup analyses (e.g., by age) lead to even smaller groups, which may reduce statistical power. While the study examines only one preprocessing variation (distortion correction), it does not assess how its effects interact with other critical steps like global signal regression or motion correction. The predictive models are limited to sensation-seeking; extending the approach to multiple behavioral phenotypes would improve robustness. Also, no external dataset was used for validation, so it's unclear how generalizable the findings are across sites or populations. Finally, the R² values, while better in the DC pipeline, still reflect modest predictive power, leaving room for model improvement.

---

### Official Review · Reviewer_Jx6i · 2025-07-17
**Review of Preprocessing Variability in fMRI Predictive Modeling: Effects of Distortion Correction on Functional Connectivity-Based Predictions**

**Confidence:** 4
**Clarity Of Writing:** good
**Clinical Significance:** fair
**Methodological Novelty:** fair
**Overall Rating:** 3
**Final Rating:** 7

**Experiments And Results:**

poor

**Questions For The Authors:**

1) Do the authors believe these findings regarding preprocessing choices would generalize to different fMRI-based prediction tasks beyond sensation-seeking?
2) How might the exclusion of female participants affect the generalizability of the results? Have the authors considered evaluating gender as a factor in their analysis?
3) How reliable and credible are the self-reported sensation-seeking scores? Could measurement noise in the target variable explain part of the poor model performance?
4) Given the relatively low R² values, how confident are the authors that these models are sufficient for evaluating the effects of preprocessing pipelines? Would stronger-performing models provide more meaningful comparisons?
5) Why did the authors choose to group participants by age rather than treat age as a continuous feature in the models, especially considering the small sample sizes and high-dimensional data?
6) What is the range of the sensation-seeking scores? Could the authors clarify this to help interpret the RMSE values and the practical significance of prediction errors?

**Strengths:**

- The study looks at an important but often overlooked part of fMRI research — how preprocessing steps, like distortion correction, can change both functional connectivity features and prediction results. This is a useful contribution to ongoing discussions about making neuroimaging studies more reliable and transparent.
- In addition to comparing model performance, the study gives helpful insights into how preprocessing affects which brain regions are linked to sensation-seeking. This makes the findings useful not just for methods, but also for understanding the brain.
- The paper is well-written, easy to follow, and supported by clear and effective visualizations that help explain the key results.

**Summary Of The Paper:**

This study examines how fMRI preprocessing strategies, specifically fieldmap distortion correction, impact functional connectivity (FC) features and the performance of predictive models in resting-state fMRI studies. Focusing on predicting sensation-seeking traits, the authors compare pipelines with and without distortion correction and demonstrate that preprocessing choices lead to substantial differences in FC measures and model outcomes. The model trained on data with distortion correction achieved higher predictive accuracy compared to the non-corrected pipeline. Additionally, key predictive connections varied between pipelines, particularly in regions affected by distortion correction, such as the cerebellum, prefrontal cortex, cingulate cortex, and subcortical areas. These findings highlight the significant influence of preprocessing decisions on FC-based modeling and underscore the need to account for such variability to improve reproducibility and reliability in neuroimaging-based prediction studies.

**Weaknesses:**

- The study focuses solely on predicting sensation-seeking traits within a specific population. It is unclear whether the findings, particularly the influence of preprocessing choices, would hold for other cognitive or clinical prediction tasks using fMRI data. This limits the generalizability of the conclusions.
- The analysis seems to be restricted to male participants, raising questions about whether the findings generalize to females or more diverse populations. Moreover, the study relies on self-reported sensation-seeking scores, which may introduce bias or noise into the prediction targets, and may potentially affect the reported model performance.
- The reported performance (R² = 0.34 for DC and R² = 0.21 for NDC) suggests that the predictive models themselves are relatively weak. Comparing preprocessing pipelines using not well-performing models may not provide reliable conclusions about the true impact of preprocessing choices. Ideally, this comparison should be conducted using well-performing models to ensure more robust and interpretable results.
- The baseline results using ElasticNet (R² = -0.03 and 0.036) further highlight the weakness of the modeling pipeline. These results suggest that the models capture little to no meaningful variance, which undermines the strength of any conclusions drawn from these comparisons.
- The decision to group participants by age, rather than include age as a continuous covariate in the models, likely reduces the statistical power given the high-dimensional nature of the data and the small sample sizes per group. This methodological choice may have contributed to the weak predictive performance.
- The study does not clearly state the range of the target variable (sensation-seeking scores), which makes it difficult to interpret the scale and significance of the reported RMSE values. Providing this context would help readers better understand the practical significance of the prediction errors.

---

### Official Review · Reviewer_mmCp · 2025-07-18
**Well-executed study on preprocessing effects in FC-based prediction**

**Confidence:** 4
**Clarity Of Writing:** excellent
**Clinical Significance:** excellent
**Methodological Novelty:** excellent
**Overall Rating:** 8

**Experiments And Results:**

great

**Questions For The Authors:**

Can you provide statistical justification for excluding females beyond sample size considerations? A sensitivity analysis including both genders would strengthen generalizability claims.

Will preprocessing scripts and derived FC matrices be made publicly available to enhance reproducibility?

**Strengths:**

Addresses a critical gap in understanding how preprocessing choices affect FC-based prediction pipelines, which has significant implications for reproducibility in neuroimaging.

Employs proper nested cross-validation with leave-one-subject-out testing, age-specific modeling approach, and systematic comparison of matched preprocessing pipelines.

Provides clear spatial correspondence between voxel displacement maps (Figure 3) and FC changes (Figure 4), offering biological plausibility for the observed effects.

The noise injection experiment (Figure 8) effectively demonstrates model vulnerability to FC variations, supporting the main findings.

Well-designed figures effectively communicate the relationship between distortion patterns and predictive connections.

**Summary Of The Paper:**

This study investigates how fieldmap distortion correction affects functional connectivity (FC) measures and subsequent machine learning prediction of sensation-seeking traits. The authors compare two preprocessing pipelines—with distortion correction (DC) and without (NDC)—on 128 male participants from the LEMON dataset. Using nested cross-validation with age-specific modeling, they demonstrate that DC achieves higher prediction performance (R² = 0.34) compared to NDC (R² = 0.21). The study reveals that brain regions most affected by distortion correction (cerebellum, prefrontal cortex, subcortical areas) overlap with key predictive connections for sensation-seeking. A sensitivity analysis shows that small variations in FC features can substantially impact model performance.

**Weaknesses:**

Male-only analysis (N=128) excludes 74 available females without adequate statistical justification beyond "larger sample size." This significantly limits the broader applicability of findings.

No code availability mentioned, insufficient detail for full replication of feature selection procedures, and proprietary fieldmap data limits reproducibility.